# Multi-Prompt Alignment for Multi-Source Unsupervised Domain Adaptation

## Abstract

Most existing methods for multi-source unsupervised domain adaptation (UDA) rely on a common feature encoder to extract domain-invariant features. However, learning such an encoder involves updating the parameters of the entire network, which makes the optimization computationally expensive, particularly when coupled with min-max objectives. Inspired by recent advances in prompt learning that adapts high-capacity deep models for downstream tasks in a computationally economic way, we introduce **M**ulti-**P**rompt **A**lignment (MPA), a simple yet efficient two-stage framework for multi-source UDA. Given a source and target domain pair, MPA first trains an individual prompt to minimize the domain gap through a contrastive loss, while tuning only a small set of parameters. Then, MPA derives a low-dimensional latent space through an auto-encoding process that maximizes the agreement of multiple learned prompts. The resulting embedding further facilitates generalization to unseen domains, making MPA naturally suitable for test time adaptation. Extensive experiments show that our method achieves state-of-the-art results on popular benchmark datasets while requiring substantially fewer tunable parameters. To the best of our knowledge, we are the first to apply prompt learning to the multi-source UDA problem and our method achieves the highest reported average accuracy of 54.1% on DomainNet, the most challenging UDA dataset to date, with only 15.9M parameters trained. More importantly, we demonstrate that the learned embedding space can be easily adapted to novel unseen domains with even fewer tuned parameters.

## 1 Introduction

Deep learning has achieved remarkable progress in various computer vision tasks such as image classification (Krizhevsky et al., 2012; He et al., 2016), object detection (Ren et al., 2015; Redmon et al., 2016; Liu et al., 2016) and image segmentation (Long et al., 2015a; Chen et al., 2017). However, these success relies on high capacity models trained in a supervised manner using a massive amount of manually labeled data, which are oftentimes expensive and time-consuming to collect. Furthermore, current deep models are brittle to the presence of domain shift (Quinonero-Candela et al., 2008; Torralba & Efros, 2011; Zhang et al., 2013) in the forms of different image styles, varied lighting conditions, diverse viewpoints, *etc.*, between training and testing distributions.

Unsupervised domain adaptation (UDA) is a popular strategy that mitigates domain discrepancies through transferring knowledge learned from a well-labeled source domain to an unlabeled target domain (Pan & Yang, 2010; Csurka, 2017; Wang & Deng, 2018). While significant advances have been achieved, current approaches focus on the single source setting, where all the labeled training data share the same distribution. In practice, however, it is more common for the labeled data to be collected from multiple sources that are diverse in distribution. Naturally, one could still tackle this problem by straightforwardly combining all the data into one single source and apply off-the-shelf UDA methods. However, directly applying single source UDA methods often results in a limited performance, as domain shift also exists among different source domains.

The integration of multiple source domains for improved adaptation results on the unlabeled target domain is generally known as multi-source unsupervised domain adaptation. Inspired by the theoretical analysis of Ben-David et al. (2006), learning domain-invariant feature representations has become a prevailing paradigm for multi-source UDA. One typical approach is to jointly learn

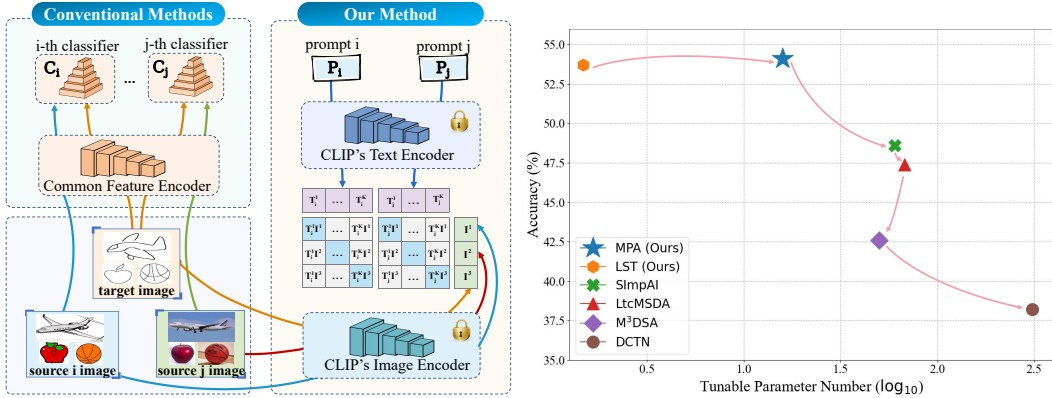

(a) Method comparison          (b) Tunable parameters vs accuracy on DomainNet

Figure 1: (a) Most conventional multi-source UDA methods use a common feature extractor with domain-specific classifier heads, while we introduce prompt learning to multi-source UDA and omit the repeated need for classifier heads. (b) MPA outperforms all other multi-source UDA methods by a large margin on the DomainNet dataset with roughly one third tunable parameters. We also introduce a **L**atent **S**pace **T**uning (LST) strategy that further reduces the trainable parameters from 15.9M (MPA) to 1.47M while still capable of achieving a high accuracy. See texts for more details.

a common feature extractor together with domain-specific feature extractors and classifier heads. Various feature distance metrics (Long et al., 2015b; Sun & Saenko, 2016; Kang et al., 2019) or domain adversarial training (Tzeng et al., 2017) can be leveraged that serves as a preliminary alignment between source and target domains, followed by different auxiliary losses carefully designed to further reduce the domain shift. While these methods offer decent results, they require optimizing the entire set of parameters in the network and hence poses a significant challenge for optimization even without the widely used min-max objective (Zhao et al., 2018; Li et al., 2018; Hoffman et al., 2018). Such problem is further amplified if we wish to apply more advanced backbones such as Vision Transformer (Dosovitskiy et al., 2021) for improved performance.

In this paper, we introduce a simple yet efficient approach for multi-source UDA without the need of retraining the entire network and the use of complicated min-max loss functions (See Figure 1 for a comparison). In particular, we build upon prompt learning (Lester et al., 2021; Wei et al., 2021) that has been designed to transfer knowledge learned from large pre-trained vision language models like CLIP (Radford et al., 2021). In prompt learning, image representations are learned contrastively with a piece of language text termed as "prompt" that describes the class of the image. While recent studies (Ge et al., 2022; Ben-David et al., 2022) suggest that learnable prompts can be used for UDA, they are restricted to the single source scenario and directly generalizing them to the multi-source setting produces limited results. Furthermore, while prompt learning provides an efficient alternative, we argue that the number of parameters needed to be tuned could be further reduced.

In light of this, we present a surprisingly easy framework, **M**ulti-**P**rompt **A**lignment (MPA), for multi-source UDA. MPA is composed of two stages, one to learn an individual prompt by tuning a small set of parameters for each source and target domain pair and one to mine the relationships among learned prompts through deriving a shared embedding space. The resulting embedding is expected to be domain-invariant and can generalize to unseen domains. More specifically, for the first stage, given a source domain and a target domain, we use CLIP as our backbone and learn one prompt tailored for such a pair. We then align all the learned prompts in a latent space of a lower dimension $d_I$. This is accomplished by a simple auto-encoder network with a reconstruction loss. Additionally, we incorporate an $\mathcal{L}_1$ loss so that the reconstructed prompts agree on the classification of target images. This is beneficial for prompts to handle situations in which the target data lies near the decision boundary. We conduct extensive experiments on multiple benchmarks and the results clearly show that our method outperforms state-of-the-art methods in the multi-source setting. In particular, on DomainNet (Peng et al., 2019), the most challenging dataset for multi-source UDA so far, MPA surpasses all state-of-the-arts methods. More importantly, as the latent space is optimized with prompts from multiple source domains, it encodes knowledge shared by different domains and could potentially generalize to unseen domains by traversing the space. Consequently, we show how surprisingly easy it is to tune the learned low-dimensional embedding for deployment in unseen

domains. Since such tuning involves no source domain data, this can be naturally extended to test time adaptation problems. In summary, our contributions are three-fold:

- We introduce **M**ulti-**P**rompt **A**lignment (MPA) for multi-source UDA. MPA takes advantage of prompt learning, thus greatly reducing the number of parameters needed for training compared with alternative methods.

- MPA learns a latent embedding space by maximizing the consensus of multiple learned prompts. Consequently we provide a methodology that is able to solve test time adaptation problems by facilitating the adaptation of the resulting low-dimensional embedding to novel unseen domains.

- MPA achieves state-of-the-art results on several popular benchmarks. Specifically, on the large-scale DomainNet dataset, MPA achieves the best reported average accuracy. Moreover, tuning the learned latent subspace offers comparable results with even fewer parameters when generalizing to unseen domains.

## 2 RELATED WORK

### 2.1 MULTI-SOURCE UNSUPERVISED DOMAIN ADAPTATION

First studied by Yang et al. (2007), multi-source UDA has drawn increasing attention in the community. Throughout the years, various methods have been studied. For example, in MDAN (Zhao et al., 2018) and DCTN (Xu et al., 2018), a discriminator applied with adversarial losses is trained so that the features from source and target domains are aligned. MFSAN (Zhu et al., 2019) calculates and aligns the maximum mean discrepancy for each source and target pair. Similarly, M$^3$SDA (Peng et al., 2019) aligns the moment distance for both target and source domains. All these methods require a shared feature extractor to obtain domain-invariant features, which is inevitably difficult to optimize and at the risk of losing semantic information as the number of domains increases. On the contrary, Rakshit et al. (2019) adopt one domain-specific encoder for each source and target pair while Zhao et al. (2020) pretrain a classifier for each source domain and then adversarially map the target images into each trained feature space. The better alignment of these methods is at the cost of significantly increased number of parameters needed for training. To overcome such a trade-off between performance and efficiency, we introduce prompt learning to multi-source UDA. Since each prompt contains substantially fewer parameters compared to a feature extractor, learning distinct prompts for each source and target pair is affordable.

### 2.2 PROMPT LEARNING

Traditionally, given a pre-trained language model, a common approach in deep learning is fine-tuning the whole model or its task-specific heads to adjust to downstream tasks. While effective, however, two main drawbacks exist. First of all, as the model size keeps increasing, pre-training and fine-tuning is becoming more and more expensive. Secondly, for each diverse task, fine-tuning needs to be repeatedly conducted. Recently, researchers have shown that learned large-scale language models can handle a wide range of downstream tasks with only a few or even no samples by pre-pending instructions to the input text (Liu et al., 2021b). Such instruction texts are called prompts. Consequently, prompts can be tuned instead of the entire network for a more efficient adaptation to downstream tasks. Originally, prompts are essentially sequences of manually designed language tokens that are mapped to an embedding space. To date, extensive research has demonstrated that training soft prompts, *i.e.*, prompts with their own parameters learned by deep models, is more effective (Li & Liang, 2021; Lester et al., 2021). The success of prompt learning in NLP has also garnered attention in the vision community that motivated the establishment of many related work. To name a few, Zhou et al. (2022) are the first to apply soft prompt learning to the image recognition task. Ju et al. (2021) explore prompt learning for efficient and lightweight video understanding. While prompting in these studies are limited to the input of text encoders, Jia et al. (2022) prepend token embeddings directly to the image patches.

## 3 METHOD

Our goal is to use multiple labeled source domains for improved performance on a target domain while only tuning a small set of parameters. To this end, we leverage prompt learning, which is an effective strategy by learning a small set of parameters to adapt a pretrained model to different downstream tasks. In the following, we first review prompt learning in CLIP in Sec. 3.1 and then elaborate in Sec. 3.2 our proposed MPA method, which is a two-stage framework that seeks agreement from multiple learned prompts and derives a joint latent space with domain-invariant knowledge.

### 3.1 AN OVERVIEW OF PROMPT LEARNING IN CLIP

CLIP consists of an image encoder and a text encoder that are jointly trained with a contrastive loss on 400M image and text pairs. The image encoder $f$, which can either be a ResNet (He et al., 2016) or a Vision Transformer (Dosovitskiy et al., 2021), maps raw images to an embedding space, and the text encoder $g$ is a Transformer (Vaswani et al., 2017) that maps an input text sequence to the same embedding space. A prompt in CLIP usually exists in the form of "a photo of [CLS]" where [CLS] is a class token that can be replaced by a certain class name. This sequence of tokens is first converted into a lower cased byte pair encoding (BPE) representation, which is essentially a unique numeric ID (Zhou et al., 2022). Then the numeric IDs are embedded to a 512 dimension vector that is further passed to the Transformer text encoder. In our work, instead of using manually crafted prompts, we train soft prompts that are directly embedded by the text encoder. Given an image $x$ and a text embedding $w_k$ for class $k \in \{1, 2, ..., K\}$, where $K$ is the total number of categories, CLIP aligns them in a contrastive manner so that:

$$p(y = k|\boldsymbol{x}) = \frac{\exp(<\boldsymbol{w}_k, f(\boldsymbol{x}) > /T)}{\sum_{i=1}^{K} \exp(<\boldsymbol{w}_i, f(\boldsymbol{x}) > /T)} \tag{1}$$

is maximized when the input image $x$ indeed belongs to class $k$. Here $< \cdot, \cdot >$ denotes the cosine similarity and $T$ is a learnable temperature parameter.

### 3.2 MULTI-PROMPT ALIGNMENT

Let $N$ denote the total number of domains, where the first $N-1$ domains are source domains and the $N$-th domain is the target domain. For all $N-1$ source domains, both images and their labels are provided, while for the target domain, we only assume access to their images, as in the standard UDA setting. For multi-source UDA, we wish to learn a domain-invariant latent space so that the domain shift among different source domains as well as the discrepancies between all the source and target domain pairs can be minimized. Unlike current multi-source UDA methods that require a significant amount of parameters for adaption, we leverage prompt learning for alignment among different domains. In the following, we begin by introducing the prompt design, followed by learning individual prompts and multi-prompt alignment. Finally we describe how to learn a joint embedding space that has the potential of generalizing to unseen domains.

**Prompt Design.** Following (Ge et al., 2022), our prompt for multi-source UDA includes a set of class-specific context vectors $\boldsymbol{v}_i^k, i \in \{1, 2, ..., M_1\}, k \in \{1, 2, ..., K\}$ and another set of domain-specific vectors shared across all classes $\boldsymbol{d}_j^d, j \in \{1, 2, ..., M_2\}, d \in \{s, t\}$. See Figure 2 for an overview. Here, $M_1$ and $M_2$ represent the number of tokens, $K$ is the number of classes, $s$ is short for source and $t$ is short for target, resulting in a total of $2K$ categories for training. In other words, each class prompt $\boldsymbol{t}_k^d \in \mathbb{R}^{1 \times (M_1 + M_2) \times 512}$ is a concatenation of a "source prompt" and a "target prompt". Therefore, the prompt for each source and target pair can be derived as:

$$\boldsymbol{P}_i = [\boldsymbol{t}_1^s, \boldsymbol{t}_2^s, ..., \boldsymbol{t}_K^s, \boldsymbol{t}_1^t, \boldsymbol{t}_2^t, ..., \boldsymbol{t}_K^t]^\top, i \in \{1, 2, ..., N-1\}. \tag{2}$$

These prompts serve as learnable parameters that help bridging the domain gap between a source domain and the target domain through a contrastive loss, as will be introduced below.

**Learning Individual Prompts.** To apply prompt learning to multi-source UDA, we first train individual prompts for each source and target pair using the image and text encoders of CLIP. Given an image $\boldsymbol{x}^s$ sampled from the source domain $\mathcal{D}_s$ whose label is $y^*$, we optimize the prompts so that the outputs from the image and text encoder are aligned. For an image $\boldsymbol{x}^t$ from the target

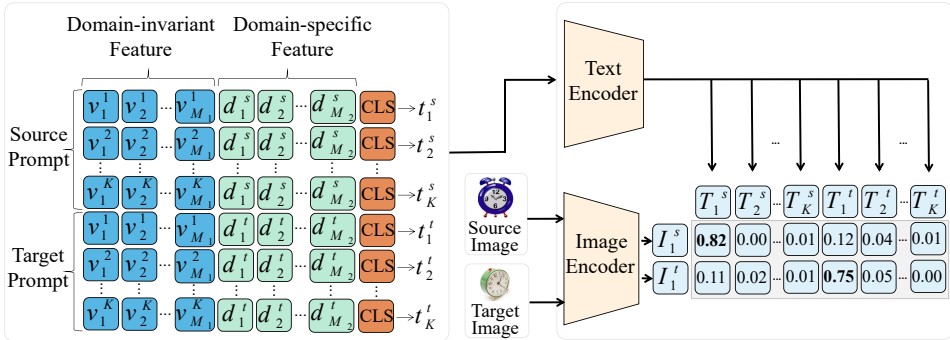

Figure 2: Each source and target pair prompt $\boldsymbol{P}_i$ is the concatenation of a "source prompt" segment and a "target prompt" segment composed of domain-invariant and domain-specific features. Therefore, the size of $\boldsymbol{P}_i$ is $\mathbb{R}^{2K \times (M_1+M_2) \times 512}$. During our prompt training stage, the text encoder and the image encoder of CLIP are both frozen.

domain $\mathcal{D}_t$ whose label is unknown, we first leverage the strong zero-shot ability of CLIP to generate static pseudo-label $\hat{y}^*$ for image-text alignment. Pseudo-labels are only generated for images whose maximum probability is larger than a fixed threshold $\tau$ using Equation 1. While more sophisticated approaches like self-training could be leveraged to generate pseudo labels (Zou et al., 2018; Liu et al., 2021a), we find that pseudo labels from CLIP are simple and effective. Finally, prompts are trained with cross-entropy loss functions and Figure 2 gives an overview of the process. More formally, for a prompt $\boldsymbol{P}_i, i \in \{1, 2, ..., N-1\}$, the objective function for optimization follows:

$$\min_{\boldsymbol{P}_i} -\frac{1}{n_s} \sum_{\boldsymbol{x}^s \sim \mathcal{D}_s} \log P(y = y^* | \boldsymbol{x}^s; \boldsymbol{P}_i) - \frac{1}{n_t} \sum_{\boldsymbol{x}^t \sim \mathcal{D}_t} \log P(y = \hat{y}^* | \boldsymbol{x}^t; \boldsymbol{P}_i). \tag{3}$$

Here, the probability $P(\cdot | \boldsymbol{x}^d; \boldsymbol{P}_i)$ of an image sample belonging to the $k$-th class is derived from a contrastive loss:

$$P(y = k | \boldsymbol{x}^d; \boldsymbol{P}_i) = \frac{\exp(< g(\boldsymbol{t}_k^d), f(\boldsymbol{x}^d) > /T)}{\sum_{d \in \{s,t\}} \sum_{i=1}^{K} \exp(< g(\boldsymbol{t}_i^d), f(\boldsymbol{x}^d) > /T)}, \tag{4}$$

where $d \in \{s, t\}$ is a domain identifier indicating where the image comes from, $T$ is a learnable temperature parameter, and $f$ and $g$ represents the image and text encoder in CLIP respectively, which are kept frozen during training. This specific design can push the prompts to learn disentangled representation of both class-invariant and class-specific semantic information to boost the performance of domain adaptation methods (Bousmalis et al., 2016; Liu et al., 2018). Once prompts are learned, the predicted label of an image $\boldsymbol{x}^t$ can be computed as:

$$\arg\max_{k} \frac{\exp(< g(\boldsymbol{t}_k^t), f(\boldsymbol{x}^t) > /T)}{\sum_{i=1}^{K} \exp(< g(\boldsymbol{t}_i^t), f(\boldsymbol{x}^t) > /T)}. \tag{5}$$

**Multi-Prompt Alignment.** Thus far, we have obtained a prompt for each source and target domain pair. However, the number of images as well as the noise level in each source domain varies, and hence these learned prompts might produce inconsistent results even for the same image. In the second stage, we aim to align predictions from different prompts and more importantly, we wish to find a domain-invariant latent space that minimizes the noise in learned prompts as well as can potentially generalize to unseen domains. To this end, we leverage auto-encoders that are trained to reconstruct the learned prompts. More formally, we use two separate auto-encoders, each consisting of a projection function $\mathbf{Proj}(\cdot)$ and a back-projection function $\mathbf{Proj}_b(\cdot)$. The learned prompts $\boldsymbol{P}_i$ are first projected into a latent subspace of a lower dimension $d_I$ by $\mathbf{Proj}(\cdot)$, followed by $\mathbf{Proj}_b(\cdot)$ projecting the vectors back into soft prompts $\hat{\boldsymbol{P}}_i$. Instead of reconstructing the whole prompt, we adjust $\boldsymbol{P}_i$ to only contain the target token segment and feed its domain-specific and domain-invariant pieces to the two auto-encoders respectively. We posit that since these two feature vectors serve different purposes, using two separate auto-encoders will help with the alignment process. The $\mathbf{Proj}(\cdot)$ function is implemented by a one-layer feed forward network while $\mathbf{Proj}_b(\cdot)$ is implemented by a two-layer nonlinear perceptron:

$$\mathbf{Proj}(\boldsymbol{P}_i) = \boldsymbol{W}_1(\boldsymbol{P}_i) + \boldsymbol{b}_1 \quad \text{and} \quad \mathbf{Proj}_b(\boldsymbol{d}_I) = \boldsymbol{W}_3(\tanh(\boldsymbol{W}_2 \boldsymbol{d}_I + \boldsymbol{b}_1)) + \boldsymbol{b}_2 \tag{6}$$

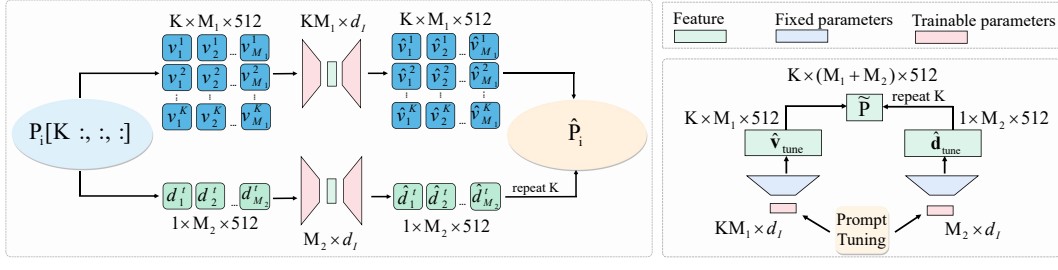

(a) Reconstruction of prompts        (b) Generalize to novel domain

Figure 3: (a) All prompts are first sliced to the size $\mathbb{R}^{K \times (M_1 + M_2) \times 512}$ containing only the target token segment. Then they are further partitioned into domain-invariant and domain-specific pieces that are projected into the same latent space and aligned by an auto-encoder structure. (b) The latent subspace learned by the auto-encoder can be utilized for generalizing to novel domain.

Here, $d_I$ represents a vector in the embedding space and we optimize a reconstruction loss

$$\mathcal{L}_{AE} = \frac{1}{N-1} \sum_{i=1}^{N-1} \|\hat{\boldsymbol{P}}_i - \boldsymbol{P}_i\|_2^2 \quad \text{where} \quad \hat{\boldsymbol{P}}_i = \mathbf{Proj}_b(\mathbf{Proj}(\boldsymbol{P}_i)). \tag{7}$$

Intuitively, one would expect a certain target domain image to be classified as the same category for all the reconstructed prompts $\hat{P}_i$. Inspired by this, we align the reconstructed prompts by introducing an additional $\mathcal{L}_1$ loss to the objective function

$$\mathcal{L}_1 = \frac{2}{(N-1) \times (N-2)} \sum_{j=1}^{N-2} \sum_{i=j+1}^{N-1} |P(y = k^t | \boldsymbol{x}_k, \hat{\boldsymbol{P}}_i) - P(y = k^t | \boldsymbol{x}_k, \hat{\boldsymbol{P}}_j)| \tag{8}$$

The overall loss function now becomes:

$$\mathcal{L} = \mathcal{L}_{CLS} + \alpha \mathcal{L}_1 + \mathcal{L}_{AE}, \tag{9}$$

where $\mathcal{L}_{CLS}$ is the cross entropy loss calculated using the reconstructed prompts $\hat{P}_i$ and static pseudo labels. Here $\alpha$ is a hyper-parameter controlling the weight of the $\mathcal{L}_1$ loss. The whole training procedure is depicted in Figure 3a. Finally, for predicting the labels of target samples, we compute the average of the output logits using each $\hat{P}_i$.

**Generalizing to Unseen Domains.** With multi-prompt alignment, we are able to derive a low-dimensional embedding space that potentially captures the relationships among different domains. With this, we introduce a **L**atent **S**ubspace **T**uning (LST) strategy. Given a novel domain, a domain-invariant feature vector $v_{tune} \in R^{N \times M_1 \times d_I}$ with a domain-specific feature vector $d_{tune} \in R^{1 \times M_2 \times d_I}$ is randomly initialized and passed to the two learned back projection function $\mathbf{Proj}_b(\cdot)$ from MPA. Consequently, an entirely new prompt $\tilde{\boldsymbol{P}}$ can be constructed for predicting image labels on the novel domain. This is achieved by minimizing the following loss function:

$$\min_{d_{tune}, v_{tune}} -\frac{1}{n_{new}} \sum_{\boldsymbol{x}^{new} \sim \mathcal{D}_{new}} \log P(y = \hat{y}^* | \boldsymbol{x}^{new}; d_{tune}; v_{tune}). \tag{10}$$

By doing so, this further decreases the number of tunable parameters by a factor of at least $(N-1) \times \frac{d_c}{d_I}$ times (for example, on the Office-Home dataset, $d_I = 150$, $d_c = 512$ and $N = 4$) as against to MPA when adapting to novel domains and is thus much more computationally efficient. Since data from the source domains are not required in the training of LST, this is in fact a problem of test time adaptation (Kundu et al., 2020; Liang et al., 2020; Wang et al., 2021). Considering that the domain of interest is not involved in the training of the latent subspace throughout the whole process, we dub it "unseen".

## 4 EXPERIMENTS

### 4.1 EXPERIMENTAL SETUP

**Datasets and metrics.** Experiments are conducted on three popular benchmark datasets of UDA to evaluate the effectiveness of MPA, including ImageCLEF, Office-Home and DomainNet. Image-CLEF is a small-scaled dataset consisting of 1,800 images from 12 object categories in 3 different

domains: ImageNet ILSVRC 2012(I), Pascal VOC 2012(P), and Caltech-256 (C). Office-Home is a medium scaled dataset consisting of about 15,500 images from 65 categories in 4 different domains: Art, Clipart, Product and Real World. DomainNet is the largest dataset to date, consisting of about 0.6 million images from 345 categories in 6 different domains: Clipart, Infograph, Painting, Quickdraw, Real and Sketch.

We use top-1 accuracy as our evaluation metric and report results of the following settings: (1) CLIP: zero-shot CLIP on the target domain, which can be regarded as a baseline of our method. (2) Source Combined: all source domains are combined into one single domain and applied with popular single-source UDA methods. Specially in this setting, we adopt the prompting method from Zhou et al. (2022) to serve as another baseline named as "Simple Prompting". (3) Multi-Source: results reported from other multi-source UDA methods. Note that we tried to re-implement state-of-the-art methods using backbone networks from CLIP, yet most of the results are unsatisfactory and thus we only report one such attempt. Similar trends are also found in (Devillers et al., 2021; Yang et al., 2022) when transferring CLIP to pure vision tasks.

## 4.2 COMPARISON TO STATE-OF-THE-ART

The results on ImageCLEF and Office-Home are shown in Table 1. For ImageCLEF, it is obvious that MPA outperforms other methods on every task with an average accuracy of 91.7%, where there is at least a 3% increase when adapting to domain C and I. For Office-Home, MPA achieves the best results except when adapting to the domain Clipart. Nevertheless, we achieve an accuracy of 75.4% on average, which is 1.3% higher than the second best method MFSAN. It is worth noting that compared to state-of-the-art method MFSAN on both datasets, MPA only trains 0.78M and 2.36M parameters, while MFSAN has a total of 51.75M and 51.80M parameters needed for optimizing (66.3 and 21.9 times larger than ours). Furthermore, to justify whether the performance gain is from a more powerful backbone, we conduct two different sets of experiments. One is applying a simple prompt learning method (Zhou et al., 2022) to the source combined scenario; and again, as Table 1 suggests, while the Simple Prompting baseline is 1% on average better than zero shot CLIP, MPA still outperforms it with a significant margin. The other experiment is by directly switching MFSAN's ResNet50 backbone pretrained on ImageNet to CLIP's image encoder. Surprisingly, when tested on the ImageCLEF dataset, the performance even drops by a small margin of 0.3%. To conclude, both results demonstrate that CLIP's backbone is not universally better and the majority of the performance gain of our method is from its domain adaptation ability.

| | ImageCLEF | | | | Office-Home | | | | |
| --- | --- | --- | --- | --- | --- | --- | --- | --- | --- |
| | → C | → I | → P | Avg | → Ar | → Cl | → Pr | → Rw | Avg |
| **Zero-Shot** | | | | | | | | | |
| CLIP (Radford et al., 2021) | 95.1 | 87.3 | 74.0 | 85.5 | 71.5 | 50.2 | 81.3 | 82.4 | 71.4 |
| **Source Combine** | | | | | | | | | |
| DAN (Long et al., 2015b) | 93.3 | 92.2 | 77.6 | 87.7 | 68.5 | 59.4 | 79.0 | 82.5 | 72.4 |
| DANN (Ganin et al., 2016) | 93.7 | 91.8 | 77.9 | 87.8 | 68.4 | 59.1 | 79.5 | 82.7 | 72.4 |
| D-CORAL (Sun & Saenko, 2016) | 93.6 | 91.7 | 77.1 | 87.5 | 68.1 | 58.6 | 79.5 | 82.7 | 72.2 |
| DAPL* (Ge et al., 2022) | 96.0 | 89.2 | 76.0 | 87.1 | 72.8 | 51.9 | 82.6 | 83.7 | 72.8 |
| Simple Prompt* | 93.6 | 90.6 | **80.9** | 88.4 | 70.7 | 52.9 | 82.9 | 83.9 | 72.4 |
| **Multi-Source** | | | | | | | | | |
| DCTN (Xu et al., 2018) | 95.7 | 90.3 | 75.0 | 87.0 | *N.A.* | *N.A.* | *N.A.* | *N.A.* | *N.A.* |
| MDDA (Zhao et al., 2020) | *N.A.* | *N.A.* | *N.A.* | *N.A.* | 66.7 | **62.3** | 79.5 | 79.6 | 71.0 |
| SImpAI$_{50}$ (Venkat et al., 2020) | 93.3 | 91.0 | 77.5 | 87.3 | 70.8 | 56.3 | 80.2 | 81.5 | 72.2 |
| MFSAN (Zhu et al., 2019) | 95.4 | 93.6 | 79.1 | 89.4 | 72.1 | 62.0 | 80.3 | 81.8 | 74.1 |
| MFSAN+CLIP* | 96.7 | 93.0 | 77.7 | 89.1 | *N.A.* | *N.A.* | *N.A.* | *N.A.* | *N.A.* |
| **MPA** (ours) | **98.6** | **96.2** | 80.4 | **91.7** | **74.8** | 54.9 | **86.2** | **85.7** | **75.4** |

Table 1: Accuracy (%) on ImageCLEF and Office-Home. * implies that the method is based on our implementation

Table 2 shows that for DomainNet, MPA exceeds other multi-source UDA methods by more than 5%. To the best of our knowledge, this is the highest reported accuracy on this dataset so far with less than one third parameters optimized compared with most state-of-the-arts methods. Regard-

| | DomainNet | | | | | | |
|---|---|---|---|---|---|---|---|
| | → Clp | → Inf | → Pnt | → Qdr | → Rel | → Skt | Avg |
| **Zero-Shot** | | | | | | | |
| CLIP (Radford et al., 2021) | 61.3 | 42.0 | 56.1 | 10.3 | 79.3 | 54.1 | 50.5 |
| **Source Combined** | | | | | | | |
| DANN (Ganin et al., 2016) | 45.5 | 13.1 | 37.0 | 13.2 | 48.9 | 31.8 | 32.6 |
| MCD (Saito et al., 2018) | 54.3 | 22.1 | 45.7 | 7.6 | 58.4 | 43.5 | 38.5 |
| DAPL* (Ge et al., 2022) | 62.4 | 43.8 | 59.3 | 10.6 | 81.5 | 54.6 | 52.0 |
| Simple Prompt* | 63.1 | 41.2 | 57.7 | 10.0 | 75.8 | 55.8 | 50.6 |
| **Multi-Source** | | | | | | | |
| DCTN (Xu et al., 2018) | 48.6 | 23.5 | 48.4 | 7.2 | 53.5 | 47.3 | 38.2 |
| SImpAI$_{101}$ (Venkat et al., 2020) | **66.4** | 26.5 | 56.6 | 18.9 | 68.0 | 55.5 | 48.6 |
| M$^3$SDA-$\beta$ (Peng et al., 2019) | 58.6 | 26.0 | 52.3 | 6.3 | 62.7 | 49.5 | 42.6 |
| LtC-MSDA (Wang et al., 2020) | 63.1 | 28.7 | 56.1 | 16.3 | 66.1 | 53.8 | 47.4 |
| T-SVDNet (Li et al., 2021) | 66.1 | 25.0 | 54.3 | 16.5 | 65.4 | 54.6 | 47.0 |
| PFSA (Fu et al., 2021) | 64.5 | 29.2 | 57.6 | **17.2** | 67.2 | 55.1 | 48.5 |
| PTMDA (Ren et al., 2022) | 66.0 | 28.5 | 58.4 | 13.0 | 63.0 | 54.1 | 47.2 |
| **MPA** (ours) | 65.2 | **47.3** | **62.0** | 10.2 | **82.0** | **57.9** | **54.1** |

Table 2: Accuracy (%) on DomainNet. * implies that the method is based on our implementation

ing individual adaptations, MPA achieves best results on most of the adaptation tasks but performs mediocre on the Quickdraw domain. Interestingly, the result is even a little worse than CLIP. We hypothesize that this is because of the large domain gap between Quickdraw and other domains. While CLIP exhibits strong performance on the DomainNet dataset, its performance on ImageCLEF and OfficeHome dataset is limited. On the contrary, MPA consistently achieves decent results regardless of the dataset assessed on. As a result, for all three datasets, MPA surpasses CLIP by 6.2%, 4.0% and 3.6% respectively. More importantly, the Simple Prompt baseline reached an average accuracy of 50.6%, 3.5% lower than MPA. All the above results further demonstrate the success of our strategy.

**Effectiveness of Generalizing to Unseen Domains**    Inspired by recent research that shows the generalization ability of large language models when adapting to unseen tasks presented with language prompts (Sanh et al., 2022), we are interested in whether MPA also possess such ability. Results from Table 1 and Table 2 are already strong indication of the success of MPA under conventional multi-source UDA setting. Now, we would like to investigate a more efficient framework, the LST strategy as mentioned in Section 3.2, for test-time adaptation tasks.

We first present a concrete example that simulates a test-time adaptation scenario. On the Office-Home dataset, LST can be conducted by training MPA using Clipart and Product as source domains and Art as target domain, while fine-tuning the latent subspace on the unseen Real World domain. Applying this to all domains from the Office-Home and DomainNet dataset produces results shown in Table 3, where compared with MPA, the performance only dropped by 0.4% on DomainNet and 1.6% on Office-Home. Nevertheless, LST achieves higher accuracy compared to most baseline methods of Table 1 and Table 2. In particular, LST is still better than CLIP with an increase of 2.4% and 3.2% respectively for the two tested datasets. Notably, only a total of 0.17M and 1.47M parameters are tuned, which is a further drop in the number of parameters (0.17M *v.s.* 2.36M, 1.47M *v.s.* 15.9M in MPA). All of these are convincing evidence showing the generalization ability of MPA.

| | Office-Home | | | | DomainNet | | | | | | |
|---|---|---|---|---|---|---|---|---|---|---|---|
| | → Ar | → Cl | → Pr | → Rw | Avg | → Clp | → Inf | → Pnt | → Qdr | → Rel | → Skt | Avg |
| CLIP | 71.5 | 50.5 | 81.3 | 82.4 | 71.4 | 61.3 | 42.0 | 56.1 | 10.3 | 79.3 | 54.1 | 50.5 |
| MPA | 74.8 | 54.9 | 86.2 | 85.7 | 75.4 | 65.2 | 47.3 | 62.0 | 10.2 | 82.0 | 57.9 | 54.1 |
| LST | 72.9 | 52.2 | 84.9 | 85.0 | 73.8 | 64.6 | 46.7 | 61.6 | 9.8 | 81.2 | 57.6 | 53.6 |

Table 3: Results (%) of different approaches when generalizing to unseen domains.

**Qualitative Results**    To obtain further insights into MPA, we visualize, in Figure 4, the distributions of prediction confidence of MPA and CLIP for the four target domains on the Office-Home dataset. We see that for Art, Prouct and Real World, MPA produces highly confident predictions for the majority of samples. As for the Clipart domain where the recognition accuracy is lower than the other domains, MPA is clearly better than CLIP that produces a distribution skewed to the bottom.

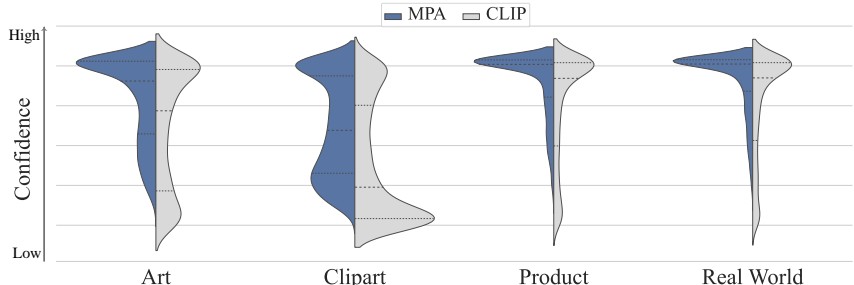

Figure 4: Visualization of the confidence density.

### 4.3 ABLATION STUDY

**Effectiveness of Prompt Alignment**    The second stage of MPA can be seen as a refinement of the prompt learned during stage one. Thus, we verify how the quality of those prompts has changed and the results are shown in Table 4. It is worth pointing out that stage one of MPA improves the accuracy of CLIP by an average of 1.2%, while the second stage improves the first stage by an average of 2.8%, highlighting that the prompt has indeed been refined.

| | A → C | A → P | A → R | C→ A | C→ P | C→ R | P→ A | P→ C | P→ R | R→ A | R→ C | R→ P | Avg |
|---|---|---|---|---|---|---|---|---|---|---|---|---|---|
| CLIP | 50.2 | 81.3 | 82.4 | 71.5 | 81.3 | 82.4 | 71.5 | 50.2 | 82.4 | 71.5 | 50.2 | 81.3 | 71.4 |
| Stage One | 52.2 | 83.5 | 82.1 | 72.8 | 83.6 | 82.8 | 73.3 | 52.7 | 82.4 | 72.0 | 51.6 | 82.1 | 72.6 |
| Stage Two | **54.1** | **85.9** | **85.2** | **74.3** | **86.0** | **85.3** | **74.6** | **54.1** | **85.3** | **74.3** | **54.2** | **86.0** | **75.4** |

Table 4: Performance (%) of individual prompts on Office-Home dataset.

**Effectiveness of loss function**    To validate the effectiveness of our objective function Equation 9, two variants are evaluated: one without the $\mathcal{L}_{AE}$ loss and one without the $\mathcal{L}_1$ loss. Table 5a shows that the removal of either piece will result in a performance degradation of about 0.7%. In particular, $\mathcal{L}_1$ loss exhibits a larger impact on the more difficult Clipart domain, where the accuracy dropped by a large margin of 1.5% without the $\mathcal{L}_1$ loss, demonstrating the effectiveness of enforcing consistent predictions on different prompts.

| $\mathcal{L}_1$ | $\mathcal{L}_{AE}$ | → Ar | → Cl | → Pr | → Rw | Avg |
|---|---|---|---|---|---|---|
| ✗ | ✓ | **74.8** | 53.4 | 85.2 | 85.3 | 74.7 |
| ✓ | ✗ | 74.0 | 53.8 | 85.5 | 85.4 | 74.7 |
| ✓ | ✓ | **74.8** | **54.9** | **86.2** | **85.7** | **75.4** |

| # of AE | → Ar | → Cl | → Pr | → Rw | Avg |
|---|---|---|---|---|---|
| Zero | **75.0** | 53.4 | 85.7 | 85.4 | 74.9 |
| One | 74.6 | 54.3 | 85.9 | 85.6 | 75.1 |
| Two | 74.8 | **54.9** | **86.2** | 85.7 | **75.4** |

| (a) Effectiveness of loss function. | (b) Effectiveness of auto-encoders. |
|---|---|

Table 5: Ablation studies.

**Effectiveness of auto-encoders**    In addition to the objective function, we also tested the effectiveness of utilizing the auto-encoder structure in MPA. For details, zero auto-encoders means we completely discarded the auto-encoder structure and one auto-encoder is for testing the necessity of using two separate auto-encoders for reconstructing domain-specific and domain-invariant prompt. Results from Table 5b show that in general, our design of the auto-encoder structure is beneficial to the overall performance. Furthermore, we find that another benefit of applying the auto-encoder structure is that it helps stabilizing the training process.

## 5 CONCLUSION

In this paper, we introduced prompt learning to multi-source UDA and proposed a simple MPA scheme to align the source and target domains. MPA is composed of two stages. The first stage is to train individual source and target pair prompts and the second stage is to align them by auto-encoder structures. Extensive experiments showed that MPA achieved better results on various multi-source UDA tasks with substantially fewer parameters tuned. Moreover, a latent subspace tuning strategy was introduced for generalization to novel unseen domains that further dropped the number of parameters needed for training and can be applied to test-time adaptation tasks. Hopefully, our work can inspire future work on applying prompt learning to transfer learning.

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

# A APPENDIX

## A.1 IMPLEMENTATION DETAILS

For fair comparisons, we adopt a ResNet50 as our backbone on ImageCLEF and Office-Home and a ResNet101 on DomainNet. The weights for ResNet50 and ResNet101 are from CLIP and frozen through our experiments. The prompts and auto-encoders are trained using the mini-batch SGD optimizer with a learning rate of 0.003 and 0.005 while the learned subspace is tuned with 0.0005 learning rate. We use a batch size of 32 and adopt a cosine learning rate scheduler. For the hyper-parameters of stage one, token length $M_1$ and $M_2$ are both set to 16. Pseudo-label threshold $\tau$ is set to 0.4 for producing reliable labels. As for stage two, $\alpha$ in Equation 9 is set to 500. Ablation studies on hyper-parameter selection is presented in the following section. The weight matrix $W_2$ of the back projection function in Equation 6 has a size of $\mathbb{R}^{384 \times d_I}$, where $d_I$ is 100 for ImageCLEF, 150 for OfficeHome and 250 for DomainNet. Therefore, for generalizing to novel domains, only 0.02M, 0.17M and 1.47M parameters are tuned respectively.

## A.2 HYPER-PARAMETER SELECTION

Experimental results for ablation studies on hyper-parameter selections are reported in Table 6.

| $\tau$ | $\rightarrow$ **Ar** | $\rightarrow$ **Cl** | $\rightarrow$ **Pr** | $\rightarrow$ **Rw** | **Avg** |
|---|---|---|---|---|---|
| 0.3 | 74.5 | **55.0** | 86.1 | **85.9** | **75.4** |
| 0.6 | 74.6 | 54.9 | 85.9 | 85.3 | 75.2 |
| 0.8 | 74.0 | 54.2 | 85.2 | 85.5 | 74.7 |
| 0.4 (reported) | **74.8** | 54.9 | **86.2** | 85.7 | **75.4** |

| **Token length** | $\rightarrow$ **Ar** | $\rightarrow$ **Cl** | $\rightarrow$ **Pr** | $\rightarrow$ **Rw** | **Avg** |
|---|---|---|---|---|---|
| $M_1 = M_2 = 8$ | 74.3 | 54.9 | 85.8 | 85.3 | 75.1 |
| $M_1 = M_2 = 12$ | 74.6 | 54.3 | 85.9 | 85.6 | 75.2 |
| $M_1 = M_2 = 20$ | **74.8** | **55.2** | **86.3** | **86.0** | **75.6** |
| $M_1 = M_2 = 16$ (reported) | **74.8** | 54.9 | 86.2 | 85.7 | 75.4 |

(a) Ablation on pseudo-label threshold $\tau$.      (b) Ablation on token lengths $M_1$ and $\mathcal{M}_2$.

| $\alpha$ | $\rightarrow$ **Ar** | $\rightarrow$ **Cl** | $\rightarrow$ **Pr** | $\rightarrow$ **Rw** | **Avg** |
|---|---|---|---|---|---|
| 1 | 74.4 | 53.7 | 84.9 | 85.6 | 74.7 |
| 10 | 74.5 | 54.1 | 85.7 | 86.0 | 75.1 |
| 100 | 74.7 | 54.5 | 85.5 | 85.6 | 75.1 |
| 1000 | 74.4 | **55.0** | **86.3** | **86.0** | 75.4 |
| 500 (reported) | **74.8** | 54.9 | 86.2 | 85.7 | **75.4** |

(c) Ablation on $\alpha$.

Table 6: Ablation studies on hyper-parameter selection.

For both pseudo-label threshold $\tau$ and prompt token length $M_1, M_2$, three different choices $\tau \in \{0.3, 0.6, 0.8\}$ and $M_1, M_2 \in \{8, 12, 20\}$ (for simplicity we are setting $M_1$ and $M_2$ to be equal) are examined. As $\tau$ increases, while the quality of the pseudo labels gets higher, fewer images will be fed into the model, and Table 6a suggests that doing so hurts the overall performance. On the contrary, shown in Table 6b, the general trend for prompt token length is that the longer the prompt, the better the performance. Consequently, we choose $\tau = 0.4$ and $M_1 = M_2 = 16$ to balance the trade-off between performance and efficiency. For $\alpha$ in Equation 9, we examined four different choices $\alpha \in \{1, 10, 100, 1000\}$ and the main reason we chose 500 for $\alpha$ is to balance all losses (in this case $\mathcal{L}_1$) to be of the same order of magnitude. Our experimental results from Table 6c also support such motivation.

## A.3 ABLATION ON PROMPT DESIGN

We also conduct experiments on ablating the necessity of using class and domain specific tokens in our prompt design.

| | $\rightarrow$ **Ar** | $\rightarrow$ **Cl** | $\rightarrow$ **Pr** | $\rightarrow$ **Rw** | **Avg** |
|---|---|---|---|---|---|
| Only class specific tokens | 74.2 | 53.0 | 85.2 | 85.3 | 74.4 |
| Class specific and domain specific tokens (reported) | **74.8** | **54.9** | **86.2** | **85.7** | **75.4** |

Table 7: Ablation on prompt design.

Note that for a fair comparison, when excluding domain specific tokens in the prompt design, the total number of tokens remains 32, *i.e.*, $\mathbf{M}_1 = 32, \mathbf{M}_2 = 0$. Still, results from Table 7 demonstrates that the reported prompt design exhibits a significant performance gain.

## A.4 SIMILARITY BETWEEN RECONSTRUCTED PROMPTS

As is observed in Table 4, the reconstructed prompts of different domains achieve almost the same results on the target domain and this is a further indication of the success of our alignment strategy. To justify such statement, the reconstructed prompts are tested without incorporating $\mathcal{L}_1$ loss in the objective function and the results are shown in Table 8

| Target \ Source | Ar | Cl | Pr | Rw | Std |
|---|---|---|---|---|---|
| **Ar** | - | 73.5 | 74.1 | 74.3 | 0.34 |
| **Cl** | 53.4 | - | 52.6 | 52.9 | 0.65 |
| **Pr** | 85.4 | 82.6 | - | 83.9 | 1.14 |
| **Rw** | 84.2 | 84.6 | 83.5 | - | 0.45 |

(a) Without $\mathcal{L}_1$ loss

| Target \ Source | Ar | Cl | Pr | Rw | Std |
|---|---|---|---|---|---|
| **Ar** | - | 74.3 | 74.6 | 74.3 | 0.14 |
| **Cl** | 54.1 | - | 54.1 | 54.2 | 0.05 |
| **Pr** | 85.9 | 86.0 | - | 86.0 | 0.05 |
| **Rw** | 85.2 | 85.3 | 85.3 | - | 0.05 |

(b) With $\mathcal{L}_1$ loss

Table 8: Comparison of performance (%) of individual prompts with and without $\mathcal{L}_1$ loss.

By comparing the standard deviation on the same target domain, it is clear that, without the $\mathcal{L}_1$ loss, the reconstructed prompts achieve different results on the target domain.

## A.5 MPA WITH LIMITED DATA

Since prompt learning is also well-known for its strong few-shot ability, a limited data setting is designed where only a small portion of the data is used for training MPA while still testing it on the full dataset to assess whether our method has inherited such ability. We think that this framework is also practically meaningful as access to data might be limited in real-life scenario due to privacy, security and storage concerns (Ahmed et al., 2021). Specifically, no more than 50 images of each category are sampled using two distinct sampling strategies. The most direct one is by complete random sampling. On the contrary, strategy two selects high quality images by utilizing CLIP's zero shot inference ability similar to our pseudo label generating process.

Such setting is only experimented on the DomainNet dataset due to its large data scale. Despite the fact that the total number of data was reduced from 0.6M to roughly 0.1M, results from Table 9 show that MPA is still capable of achieving a surprisingly good result where the recognition accuracy only dropped by an average of 0.7% compared with full data training and is still 2.9% better than our baseline CLIP. We also want to point out that the quality of input images seems to be irrelevant to the performance, as results from the two sampling strategies don't vary much.

| | → Clp | → Inf | → Pnt | → Qdr | → Rel | → Skt | Avg |
|---|---|---|---|---|---|---|---|
| CLIP | 61.3 | 42.0 | 56.1 | 10.3 | 79.3 | 54.1 | 50.5 |
| Random Sampled Data | 64.1 | 46.9 | 60.5 | 9.6 | 81.6 | 57.3 | 53.3 |
| CLIP Sampled Data | 64.2 | 47.3 | 60.9 | 9.6 | 81.6 | 57.6 | 53.5 |
| All Data | **65.2** | **47.3** | **62.0** | **10.2** | **82.0** | **57.9** | **54.1** |

Table 9: MPA with limited data on the DomainNet dataset

