# OpenReview forum: "Multi-Prompt Alignment for Multi-source Unsupervised Domain Adaptation"
_ICLR.cc/2023/Conference — Submitted to ICLR 2023_

### Official Review · Reviewer_iJo5 · 2022-10-23

**Confidence:** 4
**Correctness:** 4
**Technical Novelty And Significance:** 2
**Empirical Novelty And Significance:** 3
**Recommendation:** 5

**Clarity, Quality, Novelty And Reproducibility:**

The paper is written well; the presentation is clear and easy to understand; the novelty is a little bit weak; the experimental settings are presented and the results may be reproduced based on the details given in the manuscript.

**Strength And Weaknesses:**


++ The proposed method based on CLIP can address the multi-source UDA problems by prompt learning which is more efficient during model training.
++ The proposed method can generalise to unseen domains and the parameters to learn can be further reduced.
++ The experimental results on three benchmark datasets show competitive performance compared with others. The ablation study also validates the effectiveness of different components of the proposed approach.

-- It seems the threshold for pseudo-label selection is a very important hyper-parameter; the author should discuss how the value affects the performance.
-- The proposed method uses pre-trained CLIP as the backbone. Since CLIP is pre-trained on a different dataset from other comparative models pre-trained on ImageNet. How to justify the performance gain is not from a more powerful pre-trained model when compared with other multi-source UDA approaches? A simple baseline can be "source combine" + "simple prompt learning".
-- The authors claim the proposed approach can generalise to unseen domains by the introduced Latent Subspace Tuning (LST) strategy. Does it mean it can solve the domain generalisation problem? It seems pseudo labels are required in Eq.(10) which means the "unseen" domain is actually seen during training, please clarify this.
-- Although the proposed method aims for multi-source UDA, it seems that it can also solve single-source UDA problems. How it performs when compared with SOTA UDA approaches?


**Summary Of The Paper:**

The paper addresses the multi-source UDA problems by proposing the method Multi-Prompt Alignment (MPA). MPA is built on the pre-trained CLIP which can effectively encode images and texts. Compared to other existing approaches to mutli-source UDA, MPA only needs to train a small number of parameters by prompt learning.


**Summary Of The Review:**

The proposed approach is based on the pre-trained CLIP and utilises the prompt learning strategy to learn useful prompts for mutli-source UDA. However, the main concern is that it is not justified where the performance gains mainly come from, the proposed approach MPA or the pre-trained CLIP?

---

> ### Author Response · Authors · 2022-11-08
> **Response to reviewer iJo5 (2/2)**
>
> References:
>
> [1]Kaiyang Zhou, Jingkang Yang, Chen Change Loy, and Ziwei Liu. Learning to prompt for vision-language models. In IJCV, 2022
>
> [2] Dequan Wang, Evan Shelhamer, Shaoteng Liu, Bruno Olshausen, and Trevor Darrell. Tent: Fully test-time adaptation by entropy minimization. In ICLR, 2021.
>
> [3] Sachin Goyal, Mingjie Sun, Aditi Raghunathan, and Zico Kolter. Test-time adaptation via conjugate pseudo-labels. arXiv preprint arXiv:2207.09640, 202

---

> ### Author Response · Authors · 2022-11-08
> **Response to reviewer iJo5 (1/2)**
>
> We thank reviewer iJo5 for the positive comments and for providing detailed and thoughtful feedback on our work. We address all of reviewer’s concerns on the comments below:
>
> > *It seems the threshold for pseudo-label selection is a very important hyper-parameter; the author should discuss how the value affects the performance.*
>
> Indeed, the threshold is an important hyper-parameter and we apologize for not providing enough relevant analysis. Additional experiments were conducted to address this issue. Here we present results for three different choices of $\tau$:
>   - $\tau$ = 0.3;
>   - $\tau$ = 0.6;
>   - $\tau$ = 0.8;
>
>
>
> |             | Art   | Clipart   | Product   | Real World   | Avg |
> |:------------|:-----:|:---:|:---:|:-----:|--- |
> |$\tau$ = 0.3 |74.5 |55.0 |86.1 |85.9   |75.4|
> |$\tau$ = 0.6|74.6 |54.9 |85.9 |85.3   |75.2|
> |$\tau$ = 0.8|74.0 |54.2 |85.2 |85.5   |74.7|
> |$\tau$ = 0.4 (reported)|74.8 |54.9 |86.2 |85.7   |75.4|
>
> As the results suggest,  as $\tau$ increases, while the quality of the pseudo labels gets higher, fewer images will be fed into the model, thus hurting the overall performance. Consequently, 0.4 is a balancing choice.
>
> We will be adding these to the supplemental material of the revised version.
>
> ---
>
> > *How to justify the performance gain is not from a more powerful pre-trained model when compared with other multi-source UDA approaches? A simple baseline can be "source combine" + "simple prompt learning".*
>
> Thank you for your question and suggestion! We conducted experiments on the suggested  “source combine” + “simple prompt learning” baseline and adopted the prompting method from [1]. The results in the table below show that while the baseline is 1% on average better than zero shot CLIP, MPA is still outperforming it with a significant margin. Thus, we believe that this can justify that the performance gain is not from the powerful pre-trained CLIP but from the domain adaptation ability of our method.
>
> |             | Art   | Clipart   | Product   | Real World   | Avg |
> |:------------|:-----:|:---:|:---:|:-----:|--- |
> |CLIP         |71.5 |50.2 |81.3 |82.4   |71.4|
> |Source Combined + Simple Prompt Learning|70.7 |52.9 |82.9 |83.9   |72.4|
> |MPA          |74.8 |54.9 |86.2 |85.7   |75.4|
>
> To further answer the question, we also tested MFSAN, the second best method in Table 1, with CLIP’s pretrained backbone on the ImageCLEF dataset and the results are reported in the table below. With CLIP’s  backbone, the performance actually dropped by a small margin, demonstrating that a better backbone does not necessarily lead to a better performance. We hope that this can further justify that MPA’s performance gain is not from CLIP. These experiments will be added in the revised version of our paper.
>
> |           | C   | I   | P   | Avg   |
> |:----------|:---:|:---:|:---:|:-----:|
> |CLIP       |95.1 |87.3 |74.0 |85.5   |
> |MFSAN      |95.4 |93.6 |79.1 |89.4   |
> |MFSAN+CLIP |96.7 |93.0 |77.7 |89.1   |
> |MPA        |98.6 |96.2 |80.4 |91.7   |
>
>
>
> Finally, we would like to point out that one of the major emphasis of our work is to make the training process easier and more efficient (MPA only needs about 2% total trainable parameters compared with SOTA methods on Office-Home), rather than simply gaining better performances.
>
> ---
>
> > *Can MPA solve domain generalization problems? Pseudo labels are required in Eq.(10) which means the "unseen" domain is actually seen during training, please clarify this.*
>
> Great question! We actually tried to find a universal prompt in our derived subspace so that MPA could be extended to solve domain generalization problems. Unfortunately we haven't come up with a good solution yet and this will be our future research direction. However, with the proposed LST, MPA can actually solve the test time adaptation problem [2]. This is done by traversing the learned shared embeddings, which is expected to be domain-invariant. Throughout this stage, only data from the domain of interest is needed (no need of source domain data) and we do require pseudo labels during this phase [3]. However, this domain of interest is not involved in the training of the latent subspace, therefore we name it an “unseen domain”. We will clarify this in the revised version.
>
> ---
>
> > *How it performs when compared with SOTA single source UDA approaches?*
>
> Stage one of MPA can be considered as a single source UDA approach. We have reported its performance in Table 4 and below for reference.
>
> ||A&rarr;C|A&rarr;P|A&rarr;R|C&rarr;A|C&rarr;P|C&rarr;R|P&rarr;A|P&rarr;C|P&rarr;R|R&rarr;A|R&rarr;C|R&rarr;P|Avg|
> |:-|:-:|:-:|:-:|:-:|:-:|:-:|:-:|:-:|:-:|:-:|:-:|:-:|:-:|
> |Stage One| 52.2|83.5 |82.1 |72.8 |83.6 |82.8 |73.3 |52.7 |82.4 |72.0 |51.6 |82.1 |72.6 |
>
> ---

---

> ### Author Response · Authors · 2022-11-11
> **Response to reviewer iJo5**
>
> Dear reviewer iJo5, we would be grateful if you could confirm whether our response has addressed your concerns. Please do not hesitate to let us know whether there is anything else you would like to see clarified or improved before the end of the rebuttal period.

---

> ### Author Response · Authors · 2022-11-15
> **Response to reviewer iJo5**
>
> Dear reviewer iJo5, we are still looking forward to your reply. As the end of the rebuttal period is approaching, please let us know if there are any concerns remaining.

---

### Official Review · Reviewer_rNJs · 2022-10-24

**Confidence:** 2
**Correctness:** 4
**Technical Novelty And Significance:** 3
**Empirical Novelty And Significance:** 4
**Recommendation:** 8

**Clarity, Quality, Novelty And Reproducibility:**

The paper is well written and is novel in context of multi-source domain adaptation. The paper contains enough details for reproduction.

**Strength And Weaknesses:**

**Strengths**

* The paper demonstrates how prompt engineering/learning can be performed in context of multi-source domain adaptation.
* The paper shows that the the text encoder plays an important role in obtaining higher accuracy.
* The results on the challenging domain-net dataset is impressive. The proposed method achieves state-of-the-art results with fewer trainable parameters.

**Weaknesses**

* While the paper includes ablation studies on other methods with image encoder, it is not clear (unfair comparison) if this enough to benchmark the proposed methods against the previous method.
* It is not clear how to go about deciding the hyperparams $M_1$ and $M_2$. 16 seems a little too big for the proposed method.

**Summary Of The Paper:**

This paper explores how CLIP can be used effectively for multi-source domain adaptation. In this problem setting there is a single labeled source domain dataset and multiple target domain datasets that are not labeled. Several methods have been proposed in past to address this problem, but they all have limited accuracy owing the limited abilities of ImageNet trained model (in comparison to CLIP dataset). Authors propose a simple idea to learn prompts using pseudo-labels from CLIP and propose an auto-encoder network to learn a latent space that generalizes to new domains. While not surprising, the proposed CLIP based method outperforms the existing methods. It is interesting to note that ResNet models initialized from the CLIP's image encoder do not perform as well.

**Summary Of The Review:**

Based on the strengths and weaknesses of the paper, I vote to accept the paper.

---

> ### Author Response · Authors · 2022-11-10
> **Response to reviewer rNJs**
>
> We thank reviewer rNJs for the positive comments and for providing detailed and thoughtful feedback on our work. We address all of reviewer’s concerns on the comments below:
>
> > *While the paper includes ablation studies on other methods with image encoder, it is not clear (unfair comparison) if this enough to benchmark the proposed methods against the previous method.*
>
> Great question! As a matter of fact, reviewer Ijo5 raised a similar concern and suggested a simple baseline of “source combine” + “simple prompt learning”. We followed the reviewer's idea and the results below show that while the baseline is 1% on average better than zero shot CLIP, MPA is still outperforming it with a significant margin.
>
> |             | Art   | Clipart   | Product   | Real World   | Avg |
> |:------------|:-----:|:---:|:---:|:-----:|--- |
> |CLIP         |71.5 |50.2 |81.3 |82.4   |71.4|
> |Simple Prompt|70.7 |52.9 |82.9 |83.9   |72.4|
> |MPA          |74.8 |54.9 |86.2 |85.7   |75.4|
>
> Furthermore, for a better comparison, we tested MFSAN, the second best method in Table 1, with CLIP’s pretrained backbone on the ImageCLEF dataset and the results are reported in the table below. With the initializations from CLIP, the performance actually dropped by a small margin, demonstrating that a better backbone does not necessarily lead to a better performance.
>
> |           | C   | I   | P   | Avg   |
> |:----------|:---:|:---:|:---:|:-----:|
> |CLIP       |95.1 |87.3 |74.0 |85.5   |
> |MFSAN      |95.4 |93.6 |79.1 |89.4   |
> |MFSAN+CLIP |96.7 |93.0 |77.7 |89.1   |
> |MPA        |98.6 |96.2 |80.4 |91.7   |
>
>
> Hopefully the above added experiments can provide a better understanding of our method.
>
> > *It is not clear how to go about deciding the hyperparams $M_1$ and $M_2$. 16 seems a little too big for the proposed method.*
>
> To answer your concern, we examined three different choices of $M_1$ and $M_2$ (for simplicity we are setting them to be equal):
>   - $M_1$ = $M_2$ = 8;
>   - $M_1$ = $M_2$ = 12;
>   - $M_1$ = $M_2$ = 20;
>
>
>
> |             | Art   | Clipart   | Product   | Real World   | Avg |
> |:------------|:-----:|:---:|:---:|:-----:|--- |
> |$M_1$ = $M_2$ = 8 |74.3 |54.9 |85.8 |85.3   |75.1|
> |$M_1$ = $M_2$ = 12|74.4 |54.8 |86.1 |85.5   |75.2|
> |$M_1$ = $M_2$ = 20|74.8 |55.2 |86.3 |86.0   |75.6|
> |$M_1$ = $M_2$ = 16 (reported)|74.8 |54.9 |86.2 |85.7   |75.4|
>
> The general trend here is that the longer the prompt, the better the performance, which is not that surprising. We chose 16 mainly to balance the trade-off between performance and efficiency.

---

> ### Author Response · Authors · 2022-11-15
> **Response to reviewer rNJs**
>
> Dear reviewer rNJs, we are still looking forward to your reply. As the end of the rebuttal period is approaching, please let us know if there are any concerns remaining.

---

> > ### Comment · Reviewer_rNJs · 2022-11-15
> > **Response**
> >
> > Thanks for the rebuttal!
> >
> > I have carefully read other reviewers comments and have also read the rebuttal.
> > It is indeed interesting to see that the word length does not seem to impact the performance of the model much.
> > While 8 words might still be large, it is surprising to see that the performance of the model does not degrade much.
> >
> > Thanks again for response. I retain the score of accept since the paper is a good shape for acceptance.

---

> > > ### Author Response · Authors · 2022-11-15
> > > **Response to reviewer rNJs**
> > >
> > > Dear reviewer rNJs, we appreciate the time and effort you spent on reading all the comments and rebuttals and sincerely thank you for your support!

---

### Official Review · Reviewer_xddt · 2022-10-24

**Confidence:** 5
**Correctness:** 2
**Technical Novelty And Significance:** 2
**Empirical Novelty And Significance:** 2
**Recommendation:** 5

**Clarity, Quality, Novelty And Reproducibility:**

- Implementations are not provided
- Some technical details are not very clear for reproducibility (see above comments)

**Strength And Weaknesses:**

**Strength**

* The idea of using prompt learning for domain adaptation is interesting, as prompt learning has the potential for transfer learning in the new domain under a cheaper cost
* Experimental results show improvements over the CLIP baseline
* Ablation study shows some gains for the designed modules

**Weakness**

Technical motivation and novelty
* Although applying prompt learning for domain adaptation is interesting, the motivation is not very intuitive as prompt learning is based on a highly generalized CLIP model, which already suffers less in the domain adaptation setting. This may raise two questions: 1) whether the experimental comparisons are fair (see comments below), and 2) whether the alignment really happens as CLIP already performs very well in target domains (see comments below).

* The proposed techniques in this paper are not new, e.g., extending the prompt design to multi-domains is straightforward and the loss functions (e.g., contrastive loss) are also not new. More importantly, some designs are not well motivated and lacks experimental validations (discussed below).

* For multi-prompt alignment, it's not clear how the autoencoder is needed to achieve alignment between prompts. In experimental results (Table 5 (a)), performance gain is also marginal.

* It seems the performance of using prompt learning would highly depends on the CLIP model (e.g., Qdr results in Table 2). This should be discussed in the paper.

Technical clarity
* For eq(8), it's not clear which prompts are aligned, e.g., between P_i and P_j
* d_I appears many times in the paper but the definitions are not clear, e.g., eq(6), dimensions in v_tune and d_tune.

Experimental results
* Since CLIP already has a strong generalization ability, e.g., many results in Table 1 and 2 (Zero-Shot) are already better than other existing DA methods, it's not clear whether the comparisons are fair, as CLIP has been trained with millions of images. While I understand it's not easy to perform experiments using the CLIP backbone with existing methods, the proposed training scheme also cannot validate it's DA ability but only show the transfer learning ability.
* Prompts are designed with class-specific and domain-specific ones. However, there are no experiments to validate whether it is necessary.
* Lots of sensitivity experiments are not provided, e.g., length of prompts, thresholds for pseudo-labels, /alpha in eq(9)
* The explanation of using LST for experiments is not clear (paragraph above Table 3)


**Summary Of The Paper:**

The paper proposes a method based on prompt learning for multi-source domain adaptation. By adopting a pre-trained CLIP based text and image encoders, the authors design prompts that are learnable to adapt them for source and target domains. Specifically, there are two stages. First, prompts are designed with class-specific and domain-specific ones, and are learned for each source and target pairs individually via a contrastive loss. Then, to further align between target prompts, the authors apply the autoencoder-based method for prompt reconstruction, with a loss function to ensure output predictions are similar from different prompts. In addition, the authors also show a finetuning scheme that can adapt to new domains with small finetuning parameters. Experiments are conducted on three benchmark settings, including ImageCLEF, Office-Home, and DomainNet.

**Summary Of The Review:**

Overall, using prompt learning is an interesting direction for domain adaptation. However, there can be issues by using pre-trained CLIP that already has strong generalization ability for DA. In addition, some design choices in the proposed method are not well motivated nor well validated. The authors should consider the above comments and address them carefully in the rebuttal.

---

> ### Author Response · Authors · 2022-11-08
> **Response to reviewer xddt (3/3)**
>
> >*The explanation of using LST for experiments is not clear (paragraph above Table 3)*
>
> Sorry for the confusion. LST can be seen as a strategy for test time adaptation by traversing the learned shared embedding space. The experiments conducted is a simulation of the test time adaptation scenario, where for example on the Office-Home dataset, the embedding space can be derived through Art, Clipart and Product domain and LST conducted on the Real World domain.
>
> ___
>
> References:
>
> [1] Kaiming He, Xinlei Chen, Saining Xie, Yanghao Li, Piotr Doll ́ar, and Ross Girshick. Masked au-toencoders are scalable vision learners. In CVPR, 2022
>
> [2] Dequan Wang, Evan Shelhamer, Shaoteng Liu, Bruno Olshausen, and Trevor Darrell. Tent: Fully test-time adaptation by entropy minimization. In ICLR, 2021.
>
> [3] Yunsheng Li, Lu Yuan, Yinpeng Chen, Pei Wang, and Nuno Vasconcelos. Dynamic transfer for multi-source domain adaptation. In CVPR 2021.

---

> ### Author Response · Authors · 2022-11-08
> **Response to reviewer xddt (2/3)**
>
> >*Performance of using prompt learning would highly depend on the CLIP model (Qdr results in Table 2).*
>
> We agree that the performance of MPA is somewhat related to the CLIP model. However, MPA clearly outperforms CLIP because it is able to learn additional domain-invariant knowledge. On domain I of ImageCLEF, domain Product of Office-Home and domain Painting of DomainNet for example, MPA exceeds CLIP by a large margin of 8.9%, 4.9% and 5.9% respectively. As a matter of fact, CLIP's overall performance on the ImageCLEF and Office-Home dataset is limited as suggested by Table 1. Nevertheless, MPA still has promising results regardless of how CLIP behaves.
>
> As for the domain Quickdraw, we have discussed in the paper that MPA’s mediocre performance is most likely due to the large domain gap between quickdraw and other domains. Such a trend is also found in [3].
>
> ---
>
> >*For eq(8), it's not clear which prompts are aligned, e.g., between P_i and P_j*
>
> The absolute values of the difference between ***all*** pairs of prompts’ probabilistic outputs of target domain data are aligned since for the same target domain image, they are all expected to classify it into the same category.
>
> ---
>
> >*d_I appears many times in the paper but the definitions are not clear, e.g., eq(6), dimensions in v_tune and d_tune.*
>
> Sorry for the confusion. The $\mathbf{d}_I$ in eq(6) is in bold format and represents a randomly initialized vector in the embedding space while others are scalars that represent the dimension. This will be clarified in the revised version.
>
> ---
>
> >*Prompts are designed with class-specific and domain-specific ones. However, there are experiments to validate whether it is necessary.*
>
> Thank you for pointing this out! We have conducted the experiment for validating the prompt design and the results are presented in the table below:
>
> |             | Art   | Clipart   | Product   | Real World   | Avg |
> |:------------|:-----:|:---:|:---:|:-----:|--- |
> |Only class-specific tokens |74.2 |53.0 |85.2 |85.3   |74.4|
> |Class-specific with domain-specific tokens (reported)|74.8 |54.9 |86.2 |85.7   |75.4|
>
> The results show that the incorporation of domain-specific tokens is in fact crucial to the performance of MPA. Here, for a fair comparison, the number of class-specific tokens has been raised to 32 (MPA uses 16 class-specific and 16 domain-specifc tokens). We will be adding these to the supplemental material in the revised version.
>
> ---
>
> >*Lots of sensitivity experiments are not provided, e.g., length of prompts, thresholds for pseudo-labels, /alpha in eq(9)*
>
> Sorry for not providing experiments on hyperparameter selection! They have been conducted and the results are reported below. We will also be adding these to the supplemental material in the revised version.
>
> For prompt length, we examined three different choices of $M_1$ and $M_2$ (for simplicity we are setting them to be equal):
>   - $M_1$ = $M_2$ = 8;
>   - $M_1$ = $M_2$ = 12;
>   - $M_1$ = $M_2$ = 20;
>
>
>
> |             | Art   | Clipart   | Product   | Real World   | Avg |
> |:------------|:-----:|:---:|:---:|:-----:|--- |
> |$M_1$ = $M_2$ = 8 |74.3 |54.9 |85.8 |85.3   |75.1|
> |$M_1$ = $M_2$ = 12|74.4 |54.8 |86.1 |85.5   |75.2|
> |$M_1$ = $M_2$ = 20|74.8 |55.2 |86.3 |86.0   |75.6|
> |$M_1$ = $M_2$ = 16 (reported)|74.8 |54.9 |86.2 |85.7   |75.4|
>
> The general trend here is that the longer the prompt, the better the performance, which is not that surprising. We chose 16 mainly to balance the trade-off between performance and efficiency.
>
> For pseudo-label threshold, we also examined three different choices:
>   - $\tau$ = 0.3;
>   - $\tau$ = 0.6;
>   - $\tau$ = 0.8;
>
>
>
> |             | Art   | Clipart   | Product   | Real World   | Avg |
> |:------------|:-----:|:---:|:---:|:-----:|--- |
> |$\tau$ = 0.3 |74.5 |55.0 |86.1 |85.9   |75.4|
> |$\tau$ = 0.6|74.6 |54.9 |85.9 |85.3   |75.2|
> |$\tau$ = 0.8|74.0 |54.2 |85.2 |85.5   |74.7|
> |$\tau$ = 0.4 (reported)|74.8 |54.9 |86.2 |85.7   |75.4|
>
> As $\tau$ increases, while the quality of the pseudo labels gets higher, fewer images will be fed into the model, thus hurting the overall performance. Consequently, 0.4 is a balancing choice as the results suggest.
>
> For $\alpha$ in eq(9), we examined four different choices:
>   - $\alpha$ = 1;
>   - $\alpha$ = 10;
>   - $\alpha$ = 100;
>   - $\alpha$ = 1000;
>
>
>
> |             | Art   | Clipart   | Product   | Real World   | Avg |
> |:------------|:-----:|:---:|:---:|:-----:|--- |
> |$\alpha$ = 1 |74.4 |53.7 |84.9 |85.6   |74.7|
> |$\alpha$ = 10|74.5 |54.1 |85.7 |86.0   |75.1|
> |$\alpha$ = 100|74.7 |54.5 |85.5 |85.6   |75.1|
> |$\alpha$ = 1000|74.4 |55.0 |86.3 |86.0   |75.4|
> |$\alpha$ = 500 (reported)|74.8 |54.9 |86.2 |85.7   |75.4|
>
> The main reason why we chose 500 for $\alpha$ is to balance all losses (in this case $\mathcal{L}_1$) to be of the same order of magnitude. Our experimental results also support such motivation.
>
> ---

---

> ### Author Response · Authors · 2022-11-08
> **Response to reviewer xddt (1/3)**
>
> We thank reviewer xddt for the positive comments and for providing detailed and thoughtful feedback on our work. We address all of reviewer’s concerns on the comments below:
>
> > *Many results in Table 1 and 2 (Zero-Shot) are already better than other existing DA methods and it is not clear whether the performance gain is from CLIP.*
>
> While results in Table 2 indeed show that zero-shot CLIP is better than other existing DA methods,  many results in Table 1 show the opposite. CLIP assessed on the Office-Home dataset performs worse than all but one method, and the discrepancy is further magnified on the ImageCLEF dataset. However, regardless of CLIP’s performance, our proposed MPA consistently outperforms other methods, especially on the ImageCLEF dataset where MPA has a **6.2%** performance gain w.r.t CLIP. Whether the performance gain is from CLIP is discussed in the following question. Furthermore, we would like to point out that one of the major emphasis of our work is to make the training process easier and more efficient (MPA only needs about 2% total trainable parameters compared with SOTA methods on Office-Home), rather than simply gaining better performances.
>
> ---
>
> > *The proposed training scheme cannot validate MPA’s DA ability but only show the transfer learning ability.*
>
> Thank you for your question! As a matter of fact, reviewer Ijo5 raised a similar concern and suggested a simple baseline of “source combine” + “simple prompt learning”. We followed the reviewer's idea and the results below show that while the proposed baseline is 1% on average better than zero shot CLIP, MPA still outperforms it with a significant margin and we believe that the gain can only be from MPA’s DA ability.
>
> |             | Art   | Clipart   | Product   | Real World   | Avg |
> |:------------|:-----:|:---:|:---:|:-----:|--- |
> |CLIP         |71.5 |50.2 |81.3 |82.4   |71.4|
> |Simple Prompt|70.7 |52.9 |82.9 |83.9   |72.4|
> |MPA          |74.8 |54.9 |86.2 |85.7   |75.4|
>
>
> Furthermore, we tested MFSAN, the second best method in Table 1, with CLIP’s pretrained backbone on the ImageCLEF dataset and the results are reported in the table below. With the initializations from CLIP, the performance actually dropped by a small margin, demonstrating that a better backbone does not necessarily lead to a better performance.
>
> |           | C   | I   | P   | Avg   |
> |:----------|:---:|:---:|:---:|:-----:|
> |CLIP       |95.1 |87.3 |74.0 |85.5   |
> |MFSAN      |95.4 |93.6 |79.1 |89.4   |
> |MFSAN+CLIP |96.7 |93.0 |77.7 |89.1   |
> |MPA        |98.6 |96.2 |80.4 |91.7   |
>
> Hopefully, these added experiments can justify that MPA’s performance gain is not from CLIP and validate our method's DA ability. We will include them in the revised version of our paper.
>
> ---
>
> >*Extending the prompt design to multi-source is straightforward and the loss functions are also not new.*
>
> While extending the ***prompt design*** to the multi-source scenario is easy, we would like to emphasize that extending the ***method*** is not. As is shown in Table 1 and 2, directly applying DAPL in the source combined scenario produces limited results.
>
> As for the loss functions, we believe that each of them is crucial to the success of our algorithm. The contrastive loss and the ae loss are what enable the learning of soft prompts and the auto-encoder, and the $\mathcal{L}_1$ loss can make a better alignment as discussed in the ablation study (Table 5(a)). While these aren’t new, we believe that it is also important to push simple and existent ideas to new problems and make them work rather than trying to invent new techniques. For example, before the advent of MAE [1], the idea of masked self-supervised training was already prevailing in NLP. Nevertheless, by successfully applying it on image recognition tasks, MAE is undeniably an influential work in the computer vision community.
>
> ---
> >*For multi-prompt alignment, it's not clear how the autoencoder is needed to achieve alignment between prompts. In experimental results (Table 5 (a)), performance gain is also marginal.*
>
> Indeed, the gain is marginal. However, the major purpose of using an autoencoder is not for performance gain, but for deriving the embedding space for generalizing to unseen domains. To further clarify this, the embedding learned by the auto-encoder is expected to encode domain-invariant knowledge, and by traversing it, we can adapt to domains that weren’t involved during the training stage (this is essentially LST in the paper). The process can be seen as a test-time adaptation strategy [2]. Additionally, we find that the autoencoder structure is also helpful in stabilizing the training process, which further implies that extending existent single source method to the multi source scenario is not easy.
>
> ---

---

> ### Author Response · Authors · 2022-11-11
> **Response to reviewer xddt**
>
> Dear reviewer xddt, we would be grateful if you could confirm whether our response has addressed your concerns. Please do not hesitate to let us know whether there is anything else you would like to see clarified or improved before the end of the rebuttal period.

---

> ### Author Response · Authors · 2022-11-15
> **Response to reviewer xddt**
>
> Dear reviewer xddt, we are still looking forward to your reply. As the end of the rebuttal period is approaching, please let us know if there are any concerns remaining.

---

> > ### Comment · Reviewer_xddt · 2022-11-16
> > **Response post-rebuttal**
> >
> > Dear authors,
> >
> > Thanks for the detailed response and indeed some of the concerns are addressed, e.g., parameter sensitivity, some technical details, and ablation study for different prompts. I have a few more questions based on the rebuttal:
> >
> > - For the “source combine” + “simple prompt learning” baseline: what are the implementation details, e.g., the design of prompts and how is the model trained?
> >
> > - For "MFSAN+CLIP", how are the results on Office-Home, especially on Clipart as the CLIP baseline is much worse than MFSAN (as shown in Table 1)?
> >
> > - For LST, how does auto-encoders affect results, as the authors mention that the purpose for auto-encoder is to help LST?

---

> > > ### Author Response · Authors · 2022-11-17
> > > **Response to reviewer xddt**
> > >
> > > Dear reviewer xddt,
> > >
> > > Thank you so much for your feedback on our rebuttal and we are delighted to discuss these remaining concerns.
> > >
> > > ---
> > >
> > > >*For the “source combine” + “simple prompt learning” baseline: what are the implementation details, e.g., the design of prompts and how is the model trained?*
> > >
> > > Sorry for not providing these details! We adopted the method introduced in [1] for the prompt design. Compared with MPA, the difference is that the prompt now only contains class specific tokens. In other words, here $M_1$ = 32, $M_2$ = 0 while in MPA $M_1$ = 16, $M_2$ = 16. For each target domain, we first initialize one such prompt randomly, and train it on the source domains using only contrastive loss in a supervised manner. The prompt is then believed to have learned representations for the class labels, and finally it is tested on the target domain using contrastive pairing.
> > >
> > > ---
> > >
> > > >*For "MFSAN+CLIP", how are the results on Office-Home, especially on Clipart as the CLIP baseline is much worse than MFSAN (as shown in Table 1)?*
> > >
> > > As pointed out in the paper, we did our best to re-implement state-of-the-art methods using CLIP as the backbone network, yet most of the results are unsatisfactory. Unfortunately, MFSAN + CLIP seems only to be working on the ImageCLEF dataset. Here, we report the result on the Office-Home dataset in the table below:
> > >
> > > |             | Art   | Clipart   | Product   | Real World   | Avg |
> > > |:------------|:-----:|:---:|:---:|:-----:|--- |
> > > |CLIP |71.5 |50.5 |81.3 |82.4   | 71.4 |
> > > |MFSAN |72.1 |62.0 |80.3 |81.8 | 74.1 |
> > > |MFSAN+CLIP|58.9 |41.2 |72.0 |71.0   |75.4|
> > >
> > > Clearly,  MFSAN + CLIP fails on Office-Home even though we have tried all methods or tricks we can think of to make the performance more promising. We will also make the training code and logs publicly available.
> > >
> > >
> > >
> > >
> > >
> > > ---
> > >
> > > >*For LST, how does auto-encoders affect results, as the authors mention that the purpose for auto-encoder is to help LST?*
> > >
> > > Good point! To answer this question, additional experiments have been conducted to test how auto-encoders affect LST. Specifically, if no auto-encoders are used, we can only directly initialize a prompt from the full 512 dimension space instead of initializing a vector in the found embedding subspace. The remaining procedure is then the same as LST with auto-encoder, where we train the prompt with pseudo-labels provided by CLIP using contrastive loss. The results are shown in the following table.
> > >
> > > |             | Art   | Clipart   | Product   | Real World   | Avg |
> > > |:------------|:-----:|:---:|:---:|:-----:|--- |
> > > |LST without AE |73.5 |51.9 |84.1 |84.2   | 73.4 |
> > > |LST with AE |72.9 |52.2 |84.9 |85.0   | 73.8 |
> > >
> > > In addition to performance gains as suggested by the experiments, LST tunes much fewer number of parameters when auto-encoders are used, which we believe is a strong merit. In this specific case, since $d_I=150$, the number of parameters have actually been decreased by a factor of $\frac{512}{150}$.
> > >
> > >
> > >
> > > [1] Kaiyang Zhou, Jingkang Yang, Chen Change Loy, and Ziwei Liu. Learning to prompt for vision-language models. In IJCV, 2022

---

### Official Review · Reviewer_Zi5U · 2022-10-25

**Confidence:** 4
**Correctness:** 3
**Technical Novelty And Significance:** 2
**Empirical Novelty And Significance:** 2
**Recommendation:** 6

**Clarity, Quality, Novelty And Reproducibility:**

The paper is well written in general. However, the originality of the work is limited.

**Strength And Weaknesses:**

- Strength
    1. The introduction of prompt learning into multi-source domain adaptation is interesting.
    2. The performance of the proposed method on MSDA is good compared to previous methods.
- Weaknesses
    1. The idea of prompt learning for domain adaptation is not new, and the prompt design in this paper is highly based on the method proposed by Ge et al., 2022. So the main contribution is to extend the prompt learning for domain adaptation from the single source domain to multiple source domains. The novelty is limited.
    2. How to ensure the two-stage training will achieve the desired solutions? Since the individual prompts are learned to reduce domain shift between each individual source and the target domain while the multi-prompt alignment stage aims to align multiple source domains. The learning objective is changing in the two stages. Is it possible to combine the two stages into a single one to achieve the alignment of all the domains simultaneously?
    3. Why is the auto-encoder necessary? If the two stages of prompt learning and alignment strategy can be combined into a single stage with multiple objectives for alignment and consistent prediction, is it still require the auto-encoder for reconstruction?

**Summary Of The Paper:**

This paper proposes a multi-prompt alignment method for multi-source unsupervised domain adaptation. The main idea is to first learn an individual prompt for each source-target domain pair and then mine the relationships among learned prompts through deriving a shared embedding space. The resulting embedding is expected to be domain-invariant and can be generalize to unseen domains.

**Summary Of The Review:**

In summary, the overall novelty is limited. Moreover, the design of the model is not very well justified, which requires further clarification.

---

> ### Author Response · Authors · 2022-11-08
> **Response to reviewer Zi5U**
>
> We thank reviewer Zi5U for the positive comments and for providing detailed and thoughtful feedback on our work. We address all of reviewer's concerns on the comments below:
>
> > *The idea of prompt learning for domain adaptation is not new, and the prompt design in this paper is highly based on the method proposed by Ge et al., 2022. So the main contribution is to extend the prompt learning for domain adaptation from the single source domain to multiple source domains. The novelty is limited.*
>
> We agree that the prompt design is based on the method of Ge et al. Nevertheless, we respectfully disagree that the novelty is limited. We believe that novelty isn't only about inventing new techniques, but also about pushing simple and existent ideas to new problems and make them work. The focus of Ge et al.'s work is on single source domain adaptation. While it sounds straightforward to directly extend it to the  multi-source setting, simply doing so produces limited results. As is shown in Table 1 and 2, applying their method in the source combined scenario produces performances of 86.0%, 72.8% and 52.0% on the ImageCLEF, Office-Home and DomainNet dataset, which is on average **3.4%** lower than ours. This suggests that the use of an auto-encoder structure and the design of our loss function is important, and we are the first for this attempt, which shows the significance and novelty of our work. We believe our findings are insightful for future research.
>
>
>
> ---
>
> > *The learning objective is changing in the two stages. Is it possible to combine the two stages into a single one?*
>
> Great point! Combining the two stages is actually one of our previous attempts in extending single source prompt learning methods to the multi-source setting. However, experiments show that directly combining them would make the training process unstable and therefore we decided to split the process into two stages. The goal now is to learn individual prompts in the first stage and align them in the second. In fact, we believe this is another strong evidence to support that it is difficult to directly apply prompt learning methods to the multi-source scenario. We will be working towards this direction in our future work.
>
> ---
>
> > *Why is the auto-encoder necessary?*
>
> In addition to offering performance gains, the shared embedding space for generalizing to unseen domains cannot be found without the auto-encoder structure. To further clarify this, the embedding learned by the auto-encoder is expected to encode domain-invariant knowledge, and by traversing it, we can adapt to domains that weren’t involved during the training stage (this is essentially LST in the paper). The process can be seen as a test-time adaptation strategy [1]. Additionally, we find that the auto-encoder structure is also helpful in stabilizing the training process.
>
> ---
>
> References:
>
> [1] Dequan Wang, Evan Shelhamer, Shaoteng Liu, Bruno Olshausen, and Trevor Darrell. Tent: Fully test-time adaptation by entropy minimization. In ICLR, 2021.

---

> > ### Comment · Reviewer_Zi5U · 2022-12-12
> > **After rebuttal**
> >
> > Thanks to the authors for the detailed rebuttal. After reading the rebuttal and the comments from other reviewers. Though the proposed method extends the prompt learning from the single-domain adaptation (DAPL) to the multi-domain adaptation, I still believe the overall novelty is limited since the performance gain from the additional AE module and the corresponding loss function are marginal. Compared to DAPL, the first stage in the proposed method uses static pseudo-labels, which are also generally used in previous domain adaptation methods. If the performance gain is mostly from the static pseudo-labels, the overall contribution of the paper is limited. So I keep my original recommendation.

---

> > > ### Author Response · Authors · 2022-12-12
> > > **Response to reviewer Zi5U**
> > >
> > > Dear reviewer Zi5U,
> > >
> > > Thank you for your reply! We respectfully disagree with the following comment that you made:
> > >
> > > >*I still believe the overall novelty is limited since the performance gain from the additional AE module and the corresponding loss function are marginal*
> > >
> > > Table 4 from the paper clearly shows that stage two of MPA (where we incorporated the AE module and the loss function) increases stage one by an average of 2.8\%. We strongly belive that this gain is in fact significant. We have copied the table below for reference.
> > >
> > >
> > > ||A&rarr;C|A&rarr;P|A&rarr;R|C&rarr;A|C&rarr;P|C&rarr;R|P&rarr;A|P&rarr;C|P&rarr;R|R&rarr;A|R&rarr;C|R&rarr;P|Avg|
> > > |:-|:-:|:-:|:-:|:-:|:-:|:-:|:-:|:-:|:-:|:-:|:-:|:-:|:-:|
> > > |Stage One| 52.2|83.5 |82.1 |72.8 |83.6 |82.8 |73.3 |52.7 |82.4 |72.0 |51.6 |82.1 |72.6 |
> > > |Stage Two| 54.1|85.9 |85.2 |74.3 |86.0 |85.3 |74.6 |54.1 |85.3 |74.3 |54.2 |86.0 |***75.4*** |
> > >
> > > Please let us know if there are any remaining concerns!

---

> > > > ### Comment · Reviewer_Zi5U · 2022-12-13
> > > > **Response**
> > > >
> > > > Thanks to the authors for the clarification. I carefully double-checked the results and the auto-encoder plus the proposed loss function indeed improve the performance to a certain degree. I would like to upgrade my recommendation to 6 and lean towards acceptance.

---

> > > > > ### Author Response · Authors · 2022-12-13
> > > > > **Response to reviewer Zi5U**
> > > > >
> > > > > Dear reviewer Zi5U,
> > > > >
> > > > > Glad to know that some of the concerns have been addressed! We really appreciate that you decided to raise the score. As a kind reminder, we noticed that the scores haven't been updated yet. Would you please modify it accordingly?
> > > > >
> > > > >
> > > > > Thanks again for your thoughtful reviews and helpful suggestions to help improve our paper!

---

> ### Author Response · Authors · 2022-11-11
> **Response to reviewer Zi5U**
>
> Dear reviewer Zi5U, we would be grateful if you could confirm whether our response has addressed your concerns. Please do not hesitate to let us know whether there is anything else you would like to see clarified or improved before the end of the rebuttal period.

---

> ### Author Response · Authors · 2022-11-15
> **Response to reviewer Zi5U**
>
> Dear reviewer Zi5U, we are still looking forward to your reply. As the end of the rebuttal period is approaching, please let us know if there are any concerns remaining.

---

### Official Review · Reviewer_iJht · 2022-10-26

**Confidence:** 2
**Correctness:** 3
**Technical Novelty And Significance:** 3
**Empirical Novelty And Significance:** 3
**Recommendation:** 8

**Clarity, Quality, Novelty And Reproducibility:**

- What is the difference between stage one and DAPL? Stage one (learning individual prompts) seems to be similar to DAPL.

- The effectiveness of L_CLS loss in stage 2. Table.5 (a) shows the effectiveness of objective function Equation 9. However, the results of L_CLS loss are missed.

- What does “Zero” mean in Table.5 (b)? “Zero auto-encoders means we completely discarded the auto-encoder structure” is not clear to me. If the autoencoder structure is discarded, what structure is adopted?

- The similarity between the reconstructed prompts. In Table.4, the reconstructed prompts of different domains achieve almost the same results on the target domain. Does this mean that these reconstructed prompts are the same?

**Strength And Weaknesses:**

Strengths:
1.	This paper is first to apply prompt learning to multi-source UDA problem.
2.	The proposed framework is simple and achieves state-of-the-art results on multi-source UDA benchmark.
3.	The paper is well written, easy to understand.


**Summary Of The Paper:**

This paper introduces prompt learning to multi-source UDA. A simple two-stage framework is proposed. In the first stage, individual prompts for each source and target pair are learned. In the second stage, Multi-Prompt Alignment (MPA) is proposed to align the learned prompts. The experimental results show the effectiveness of the proposed framework.

**Summary Of The Review:**

Overall, this is a good submission with strong results. I hope the author could provide more detailed explanations.

---

> ### Author Response · Authors · 2022-11-10
> **Response to reviewer iJht**
>
> We thank reviewer iJht for the positive comments and for providing detailed and thoughtful feedback on our work. We address all of reviewer's concerns with additional details on the comments below:
>
> > *What is the difference between stage one and DAPL? Stage one (learning individual prompts) seems to be similar to DAPL.*
>
> Good question! The main difference is that stage one of MPA uses ***static*** pseudo-labels generated by zero-shot CLIP while DAPL uses dynamically changing pseudo-labels. Consequently, stage one of MPA converges much faster than DAPL.
>
>  > *The effectiveness of L_CLS loss in stage 2. Table.5 (a) shows the effectiveness of objective function Equation 9. However, the results of L_CLS loss are missed.*
>
> Thank you for pointing this out. We have conducted the experiment and reported the results in the table below:
>
> |             | Art   | Clipart   | Product   | Real World   | Avg |
> |:------------|:-----:|:---:|:---:|:-----:|--- |
> |Without $\mathcal{L}_{CLS}$ |72.2 |48.4 |84.9 |83.8   | 72.3 |
> |With $\mathcal{L}_{CLS}$|74.8 |54.9 |86.2 |85.7   |75.4|
>
> The reason why we didn't include them in Table 5(a) is that our method is dependent on the generated pseudo-labels, which will be meaningless without $\mathcal{L}_{CLS}$. The above experimental results show that indeed the performance drops by a large margin when $\mathcal{L}_{CLS}$ is not included in the objective function (especially on the Clipart domain).
>
> > *What does “Zero” mean in Table.5 (b)? “Zero auto-encoders means we completely discarded the auto-encoder structure” is not clear to me. If the autoencoder structure is discarded, what structure is adopted?*
>
> Zero means that we directly align the learned prompts without using the auto-encoder. This experiment is for testing whether the auto-encoder is beneficial for the alignment process, and results show that there are some gains. In addition to performance gains, the major purpose of the auto-encoder structure is for deriving the shared embedding space for generalizing to unseen domains.
>
>
> > *The similarity between the reconstructed prompts. In Table.4, the reconstructed prompts of different domains achieve almost the same results on the target domain. Does this mean that these reconstructed prompts are the same?*
>
> Indeed, the reconstructed prompts achieve almost the same results on the target domain and we believe that this is due to the success of our alignment strategy. For example, with the $\mathcal{L}_1$ loss, all prompts are constrained to produce a similar logit for the same input image of the target domain. To justify this, we conducted an additional experiment where the $\mathcal{L}_1$ loss is not included in the objective function and tested the reconstructed prompts. The results are reported in the table below:
>
> ||A&rarr;C|A&rarr;P|A&rarr;R|C&rarr;A|C&rarr;P|C&rarr;R|P&rarr;A|P&rarr;C|P&rarr;R|R&rarr;A|R&rarr;C|R&rarr;P|Avg|
> |:-|:-:|:-:|:-:|:-:|:-:|:-:|:-:|:-:|:-:|:-:|:-:|:-:|:-:|
> |Without $\mathcal{L}_1$ loss| 53.4|85.4 |84.2 |73.5 |82.6 |84.6 |74.1 |52.6 |83.5 |74.3 |52.9 |83.9 |73.8 |
> |With $\mathcal{L}_1$ loss| 54.1|85.9 |85.2 |74.3 |86.0 |85.3 |74.6 |54.1 |85.3 |74.3 |54.2 |86.0 |74.9 |
>
> The results clearly show that without the $\mathcal{L}_1$ loss, the reconstructed prompts achieve different results on the target domain.

---

> ### Author Response · Authors · 2022-11-15
> **Response to reviewer iJht**
>
> Dear reviewer iJht, we are still looking forward to your reply. As the end of the rebuttal period is approaching, please let us know if there are any concerns remaining.

---

> > ### Comment · Reviewer_iJht · 2022-11-16
> > **Response**
> >
> > Dear authors, thank you for providing the rebuttal. I read your response carefully. My concerns have been well address. I don't have further questions at the moment.

---

> > > ### Author Response · Authors · 2022-11-16
> > > **Response to reviewer iJht**
> > >
> > > Dear reviewer iJht, we sincerely thank you for your support!

---

### Author Response · Authors · 2022-11-16
**Paper Revision**

Dear reviewers, we sincerely thank all of you for your valuable time reviewing our submission. The paper has been revised in order to clarify misunderstandings, address important concerns of the reviewers, and improve the presentation of our paper. Hopefully our revision and rebuttal can better clarify the significance of this work. Thank you again for all the insightful suggestions that helped us improve our work.

---

### Author Response · Authors · 2022-12-10
**Response to reviewers**

Dear reviewers, we sincerely thank all of you again for your valuable feedback. As the end of the discussion period is approaching, we would like to know if there are any concerns remaining. Please do not hesitate to let us know.

---

### Decision · Program_Chairs · 2023-01-20

**Decision:**

Reject

**Justification For Why Not Higher Score:**

Out of the five review reports, two are negative. AC read the paper and considered all reviewing material, and agreed that the current version is relatively lacking in technical novelty and evaluation rigor. In general, its flaws are somewhat outweighing its merits.

**Justification For Why Not Lower Score:**

N/A

**Metareview: Summary, Strengths And Weaknesses:**

This paper introduces prompt learning to multi-source unsupervised domain adaptation (UDA). A two-stage framework is proposed based on the pre-trained CLIP model. In the first stage, individual prompts for each source and target pair are learned via a contrastive loss. In the second stage, Multi-Prompt Alignment (MPA) is proposed to align the learned prompts, which applies the autoencoder-based method for prompt reconstruction, with a loss function to ensure that output predictions are similar from different prompts. In addition, the authors also introduce a Latent Subspace Tuning (LST) scheme that can further decrease the number of tunable parameters for new domains. Experiments are conducted on three benchmark settings, including ImageCLEF, Office-Home, and DomainNet.

Even after the discussion phase, there exists disagreement among the five reviewers --- three positive votes, and two negative votes. AC read the paper, and considered all reviews, author responses, and the discussions. The paper has some merits, such as the interesting idea of using prompt learning in domain adaptation and state-of-the-art results on multi-source UDA benchmark. However, there are still several reservations held both by the negative reviewers and further by the AC, which prevent from direct acceptance:

1. Some reviewers still have some concerns about the technical novelty and contribution. The idea of prompt learning for domain adaptation is not new, and the prompt design in this paper in particular the prompt learning of the first stage are highly based on the method proposed by (Ge et al., 2022). So, the main contribution is to extend the prompt learning for domain adaptation from the single source domain to multiple source domains, which may be limited and less novel. Besides, the insights and motivations of the method design are not described or validated clearly enough, as concerned by the reviewers.

2. The authors claim the proposed approach can generalize to unseen domains by the introduced LST strategy. However, data and pseudo labels are still needed to further tune some parameters. It is unclear whether such prompt tuning provides good insight, compared to other clearly practical usage, e.g., test-time adaptation with prompting.

3. Some reviewers had concerns that the success of the method may depend on the performance of the pre-trained model and that using a more powerful CLIP model may be unfair to the compared methods. The authors have made some efforts to clarify this, but the reviewers are not fully convinced finally.

Given the above reservations, AC could not accept the paper for now but encourage the authors to fully revise the paper and strengthen their work.